# Prognosis of Stage I Endometrial Cancer According to the FIGO 2023 Classification Taking into Account Molecular Changes

**DOI:** 10.3390/cancers16020390

**Published:** 2024-01-17

**Authors:** Bozena Dobrzycka, Katarzyna Maria Terlikowska, Oksana Kowalczuk, Jacek Niklinski, Maciej Kinalski, Sławomir Jerzy Terlikowski

**Affiliations:** 1Department of Gynecology and Practical Obstetrics, Medical University of Bialystok, 15-295 Bialystok, Poland; bozena.dobrzycka@umb.edu.pl; 2Department of Food Biotechnology, Medical University of Bialystok, 15-295 Bialystok, Poland; katarzyna.terlikowska@umb.edu.pl; 3Department of Clinical Molecular Biology, Medical University of Bialystok, 15-269 Bialystok, Poland; oksana.kowalczuk@umb.edu.pl (O.K.); jacek.niklinski@umb.edu.pl (J.N.); 4Department of Gynecology and Obstetrics, Independent Public Healthcare Facility Regional Complex Jan Sniadecki Hospital in Bialystok, 15-595 Bialystok, Poland; mkinalski@wp.pl; 5Department of Obstetrics, Gynecology and Maternity Care, Medical University of Bialystok, 15-295 Bialystok, Poland

**Keywords:** 2023 FIGO endometrial cancer staging, molecular classification, prognostic factors

## Abstract

**Simple Summary:**

Optimum risk stratification in the early stages of endometrial cancer combines molecular and clinicopathological features. The purpose of the study was to determine the prognostic value of molecular classification and traditional pathological factors in a sample group of patients with stage I endometrial cancer according to the FIGO 2023 criteria to achieve a more personalized approach to patient care and treatment. The immunohistochemistry for p53 and mismatch repair (MMR) proteins, and DNA sequencing for *POLE* exonuclease domain and clinicopathological parameters, including outcomes in 139 patients, were analyzed. Our research studies confirm that molecular category corresponds to a different prognosis in clinical stage I EC according to the new 2023 FIGO classification, with *POLE*mut cases presenting the best outcomes and p53abn cases showing the worst outcomes. Beyond the previous routine clinicopathological assessment, the new EC staging system represents an important step toward improving our ability to stratify IC stage EC risk.

**Abstract:**

Optimum risk stratification in an early stage of endometrial cancer (EC) combines molecular and clinicopathological features. The purpose of the study was to determine the prognostic value of molecular classification and traditional pathological factors in a sample group of patients with stage I EC according to the FIGO 2023 criteria, to achieve a more personalized approach to patient care and treatment. The immunohistochemistry for p53 and mismatch repair (MMR) proteins, and DNA sequencing for *POLE* exonuclease domain and clinicopathological parameters, including disease disease-free survival (DFS) and overall survival (OS) in 139 patients, were analyzed. It has been shown that the independent recurrence risk factors are stage IC (*p* < 0.001), aggressive histological types EC (*p* < 0.001), and the presence of p53abn protein immunoexpression (*p* = 0.009). Stage IC (*p* = 0.018), aggressive histological types EC (*p* = 0.025) and the presence of p53abn protein immunoexpression (*p* = 0.010) were all significantly associated with lower 5-year OS rates. Our research studies confirm that the molecular category corresponds to a different prognosis in clinical stage I EC according to the new 2023 FIGO classification, with *POLE*mut cases presenting the best outcomes and p53abn cases showing the worst outcomes. Beyond the previous routine clinicopathological assessment, the new EC staging system represents an important step toward improving our ability to stratify IC stage EC risk.

## 1. Introduction

According to the Global Cancer Statistics report from 2020, endometrial cancer (EC) ranks as the sixth most prevalent cancer among women worldwide. In that year alone, there were 417,000 newly diagnosed cases. Over the past three decades, there has been a notable 132% increase in the incidence of this cancer, largely attributed to rising obesity rates and an aging demographic [1]. Endometrial cancer often manifests initially as postmenopausal bleeding, yet such bleeding is indicative of underlying endometrial cancer in only 5% to 10% of instances. Over 50% of the women diagnosed with endometrial cancer are detected at an early stage, making their condition more manageable and often treatable, primarily through surgical intervention. The prognosis of EC, as well as the risk of its recurrence, is shaped by several clinicopathologic factors. These include the patient’s age, which plays a critical role in determining outcomes. Additionally, the stage of the cancer is a key determinant, as it reflects the extent of the disease’s progression. The histologic subtype of the cancer also significantly impacts its prognosis. Another vital factor is the tumor grade, which indicates the aggressiveness of the cancer. Finally, the presence or absence of lymphovascular space invasion (LVSI) is crucial in assessing the likelihood of cancer recurrence. Each of these factors individually and collectively contributes to the overall prognosis and recurrence risk in EC cases [2].

Since the publication of the last update of the FIGO classification for EC in 2009, there has been significant progress in understanding the molecular pathomechanisms of its development [3,4]. Today, much more information is available to better define the molecular changes in specific types of EC. Numerous studies on new treatments and the prognostic factors that correlate with previously-recognized clinicopathologic factors have also been published. In 2023, the FIGO Committee on Women’s Cancer introduced a revised and detailed classification system for EC. This new taxonomy delineates EC into several distinct categories based on their histopathological characteristics and aggressiveness. These categories include endometrioid carcinoma (EEC), which is further subdivided into low grade (G1 and G2) and high grade (G3), serous carcinoma (SC), clear cell carcinoma (CCC), mixed carcinoma (MC), undifferentiated carcinoma (UC), and carcinosarcoma (CS). Additionally, this classification encompasses other rarer types, such as mesonephric-like carcinomas and gastrointestinal mucinous type carcinomas. In this updated framework, non-aggressive histological types are classified under low-grade endometrioid carcinomas (G1 and G2 EECs). Conversely, the aggressive histological types are now identified as high-grade endometrioid carcinomas (G3 EECs), along with SC, CCC, MC, UC, CS, and the aforementioned rare types like mesonephric-like and gastrointestinal-type mucinous carcinomas [5]. 

The new molecular and histologic staging system is based on the ESGO/ESTRO/ESP guidelines [6]. Based on the existing evidence, Berek et al. [5] defined the substages as follows. IA1 is a non-aggressive histologic type of EC Stage I (IA1), restricted to polyps or the endometrium. According to WHO criteria, IA2 is a non-aggressive histologic type of EC involving <50% of the myometrium with no or with focal LVSI. IA3 is a low-grade EC in the uterus and is associated with low-grade endometrioid ovarian disease. IB is a non-aggressive histologic class covering 50% or more of the myometrium but not having LVSI or focal LVSI. IC is an aggressive histological form with no myometrial invasion of EC, such as serous, high-grade endometrioid, clear cell, carcinosarcoma, undifferentiated, mixed, and other rare types.

The Cancer Genome Atlas (TCGA) defines optimal risk stratification in early EC as a combination of clinicopathologic factors and molecular classification. An important factor in the rapidly growing popularity of the TCGA classifier is the relative ease with which the *POLE*mut, MMRd, NSMP, and p53abn molecular subgroups can be identified. Different subclasses are discovered using IHC for p53 (p53abn EC); for example, the MSI subclass by IHC for mismatch repair (MMR) proteins (MMRd EC) and targeted sequencing of the POLE exonuclease domain (*POLE*mut EC). Malignancies without all three of these characteristics are classed as p53 wild-type (p53wt EC) or as ECs with no specific molecular profile (NSMP EC) [7,8,9].

To better reflect the heterogeneity of EC, the updated FIGO staging of EC offers various histological types and clinical staging with molecular classification in the early stages [10]. This development has resulted in an approach change from a morphology-based classification to an integrated model built around molecular and histological features [11]. Adding a molecular subtype to the staging criteria allows for better prognosis prediction. Performing molecular classification based on the assessment of *POLE*mut, MMRd, NSMP, and p53abn mutations is now recommended in all cases of EC for prognostic risk-group stratification and as factors that might influence adjuvant and systemic treatment decisions. 

The purpose of the study was to determine the prognostic value of molecular classification and traditional pathological factors in a sample group of patients with stage I EC according to the FIGO 2023 criteria, to achieve more personalized approach to patient care and treatment.

## 2. Materials and Methods

### 2.1. Patients and Clinical Samples

This study encompassed 167 patients diagnosed with FIGO stage I EC, as classified by the 1988 FIGO guidelines which were revised in 2009. These individuals were recruited from 2006 to 2018 and participated in the research conducted at the Department of Gynecology and Obstetrics, Jan Sniadecki Hospital, part of the Independent Public Healthcare Facility Regional Complex in Bialystok, Poland. The study’s protocol received Bioethics Committee approval number R-I-003/177/2004 in Poland at the Medical University of Bialystok.

Patient inclusion involved obtaining informed consent after briefing them about the study’s objectives. Surgical treatment, conforming to FIGO standards, was administered to all participants. This involved a hysterectomy with bilateral salpingo-oophorectomy for G1/G2 cases and an additional pelvic and para-aortic lymphadenectomy for G3 cases. Preoperative evaluations included standard blood tests, chest radiography, and abdominal ultrasounds encompassing the pelvic region. Advanced imaging techniques like MRI, CT, and PET-CT scans were also employed in certain cases. It is noteworthy that none of the patients received postoperative treatment. 

Histopathological assessments adhered to the guidelines and classification system of the World Health Organization (WHO). For accurate clinical staging and comprehensive evaluation, representative tissue samples were procured. These samples underwent hematoxylin and eosin (H&E) staining. Following the staining procedure, the tissue samples were subject to detailed examination under a light microscope. This meticulous microscopic analysis was instrumental in accurately assessing several crucial pathological parameters. These parameters included the extent of myometrial infiltration by the tumor cells, the specific histopathological classification of the tumor, encompassing both its type and histological grade, and the evaluation of LVSI. This comprehensive microscopic scrutiny was pivotal in providing a thorough understanding of the tumor’s characteristics and its potential behavior. All morphological assessments were performed by two independent pathologists blinded to the outcomes of each case. LVSI was defined in accordance with the 3-tiered system at the tumor invasive front (none LVSI or focal—a single focus of LVSI was recognized around a tumor) [12,13]. No cancer cells were found in the lymph nodes of the patients in the study group. 

The classification and tumor staging follow the 2023 FIGO staging standard. Patients were followed up every three months for the first two years, every six months for the next three years, and finally, once a year. After the completion of initial treatment, tumor recurrence was confirmed through physical and ultrasound examination or imaging examination (MRI, CT, PET-CT).

### 2.2. Molecular Profiling

#### 2.2.1. DNA Extraction

Tissue samples for molecular analyses were collected intraoperatively. Following surgical removal, all samples were processed immediately. Then, after macroscopic visual assessment, small pieces (about 0.2–0.5 cm^3^) of tumor tissue were immediately frozen in liquid nitrogen and finally stored in freezers at −80 °C. Prior to nucleic acid extraction, cryo-sections of frozen tissue specimens were treated with H&E to stain them and were evaluated by experienced pathologists to confirm tumor cell content suitability. Following microscopic examination, only tumor samples containing at least 50% tumor cells were chosen for further processing. Other sections of the tumor were examined to select a suitable specimen when there was insufficient tissue or the presence of necrotic areas.

In the process of extracting DNA from tissue specimens, we employed an automated technique utilizing magnetic DNA extraction. This method involved the initial lysis of approximately 40–50 milligrams of tissue using a lysis buffer provided by Biomerieux, based in Lyon, France. The lysis was facilitated using a Tissue Ruptor from Qiagen, Hilden, Germany. Following lysis, the samples were subjected to a two-hour incubation at a temperature of 56 °C with the addition of Proteinase K, an enzyme crucial for the digestion of proteins. Subsequent to the digestion and deproteinization of the lysates, the nucleic acids were efficiently isolated using the EasyMag system, also from Biomerieux, Lyon, France. This isolation was achieved through the incorporation of magnetic beads, adhering to the protocol as outlined by the manufacturer. This methodology ensured a precise and efficient retrieval of the DNA from the tissue samples. 

#### 2.2.2. *POLE* Gene Mutation Analysis

*POLE* gene exons 9–14 were amplified using the standard polymerase chain reaction (PCR) and the resulting PCR products were directly sequenced for analysis. In summary, specific exons of the gene were amplified using a PCR reaction with an appropriate forward (F) and reverse (R) primer pair. The sequences of the primers used in the assay were designated with Primer3Plus (https://www.primer3plus.com (accessed on 1 January 2019)) software based on the standardized genetic sequence that represents the *POLE* gene (LRG_789, NG_033840.1). The list of primer sequences is presented in Appendix A. The PCR reaction was carried out using 20 μL of reaction mixture containing 1 × PCR buffer with 1.5 mM MgCl_2_ (final concentration), 0.8 μL of 0.2 mM Deoxynucleotide Triphosphates (dNTPs) solution, 1 μL of both forward and reverse primer 10 μM solutions, 0.1 μL of Taq DNA polymerase, and 20–50 ng of isolated DNA (all the reagents from EURx Ltd., Gdansk, Poland). The PCR conditions were as follows: 5 min denaturation at 94 °C and then 40 cycles at 94 °C for 1 min, 57 °C for 1 min, and 72 °C for 1 min, with an extension at 72 °C for 5 min.

The agarose gel electrophoresis was used to verify and separate the PCR products, which were then extracted from the agarose slides using a “PCR clean-up Gel extraction kit” (Macherey-Nagel, Duren, Germany). To perform the extraction automatically, the QIAcube machine was used (Qiagen, Hilden, Germany), following the manufacturer’s instructions.

Direct sequencing of the clean PCR products was performed bi-directionally using an automated DNA sequencer 3500 Genetic Analyzer (Applied Biosystems, Foster City, CA, USA) and the BigDye Terminator v. 3.1 cycle sequencing kit (Applied Biosystems). Nucleotide sequences were compared with human *POLE* gene reference sequence NG_033840.1 (Locus Reference Genomic Sequences Database, LRG_789) using Nucleotide BLAST software 2.13 (https://blast.ncbi.nlm.nih.gov/Blast.cgi (performed online)). All *POLE* mutations were sequenced twice at the minimum.

### 2.3. Immunohistochemistry

#### 2.3.1. Evaluation of p53 Protein Expression and Immunohistochemistry

Two pathologists selected one representative formalin-fixed paraffin-embedded (FFPE) tumor block for every single instance. Each of the 10% FFPE tumor samples was cut into four-micrometer-thick sections and stained for immunoreactivity to a monoclonal antibody (Monoclonal Mouse Anti-Human p53 Protein Clone DO-7; Agilent Dako, Denmark) [14]. 

The WHO Classification of Tumors 2020 edition defines abnormal p53 (mutation-type) staining as either strong nuclear expression in tumor cells (>80%), a total lack of protein expression in tumor cells alongside internal control, or, in rare cases, definitive cytoplasmic expression [7,15]. For p53 IHC interpretation, a three-tier system with overexpression and a complete absence of immunoexpression was used. Positive internal controls with non-neoplastic cell staining, such as lymphocytes, fibrocytes, or endothelial cells, were used. Overexpression and complete absence of p53 IHC were interpreted as abnormal/aberrant/mutation-type. A case was classified as p53abn if it had a p53 mutation as determined by IHC (0 or 2+). In contrast to the normal/wild-type pattern, the case was classified as NSMP/p53wt due to p53 expression levels (IHC +1) in between these extremes.

#### 2.3.2. Immunohistochemical Assessment and Evaluation of MMR Protein Expression

Staining for MMR proteins was performed on all specimens. It was used to stain the 10% FFPE four-micrometer-thick sections according to the manufacturer’s instructions (Agilent Dako, Glostrup, Denmark). Monoclonal antibodies specific to the mismatch repair genes were employed, including Mouse Monoclonal Anti-Human MutL Protein Homolog 1 (MLH1, Clone ES05), Mouse Monoclonal Anti-Human MutS Protein Homolog 2 (MSH2, Clone FE11), Rabbit Monoclonal Anti-Human MutS Protein Homolog 6 (MSH6, Clone EP49), and Rabbit Monoclonal Anti-Human Postmeiotic Segregation Increased 2 (PMS2, Clone EP51). All four proteins were detected in the nucleus. When nuclear staining of any of the four proteins was missing, or MLH1/PMS2 or MSH2/MSH6 were negative, mismatch repair deficiency (MMRd) was suspected. Endometrial cancer that lacked a pathogenic *POLE* variant but retained MMR protein expression and wild-type p53 IHC was classified as a “no specific molecular profile” (NSMP) variant. Upon observation of a complete loss of nuclear expression of certain MMR proteins (MLH1, MSH2, MSH6, PMS2) in carcinoma cells using IHC, the proteins are considered deficient or absent. Atypical p53 protein expression in carcinoma cells was identified as either intense and widespread nuclear staining or a total absence of staining, termed “null.” Meanwhile, faint and uneven staining was categorized as indicative of wild-type expression. For both MMR and p53 staining, stromal and inflammatory cells served as internal controls. When applicable, samples with few carcinoma cells or completely negative staining of the internal controls were discarded.

### 2.4. Statistical Analysis

The reverse Kaplan-Meier method helped to calculate the median follow-up time. The recurrence rate was calculated by excluding patients who died from causes other than EC between the date of surgery and the date of the first relapse. The period of time between the surgery date and the death from any cause was referred to as overall survival (OS). The amount of time after treatment that no sign of cancer was found was referred to as disease-free survival (DFS) time. Analyses took place successively until December 2022. To compare patient clinicopathological characteristics across molecular subgroups, we used the Chi-Square Test of Independence. To compare an ordered categorical variable and non-normally distributed continuous variables, we used the Mann-Whitney U test. The log-rank test was used to compare recurrence, OS, and DFS between groups. Multivariate regression analyses with prespecified covariates were performed, including age (<60 years vs. 60 years or older), stage (IA vs. IB and IA vs. IC), histological subtype (endometrioid endometrial cancer; EEC G1/G2 vs. ECC G3, Serous, CCC), LVSI (none vs. focal), myometrial invasion (<50% vs. ≥50%), and molecular subgroups (*POLE*mut vs. p53abn, *POLE*mut vs. MMRd, *POLE*mut vs. NSMP). Finally, the multivariable Cox regression models included the propensity score as a covariate. A p-value of less than 0.05 on both sides was considered statistically significant. Statistica 13.3PL (StatSoft, Kraków, Poland) was used for statistical analysis.

## 3. Results

During the study, 167 samples of the EC were collected from patients with stage I disease. Molecular testing was accurate for 139 patients. The EC in these patients was classified into one of the following molecular subgroups: 47 p53abn (33.8%), 29 MMRd (20.9%), 35 *POLE*mut EC (25.2%), and 28 NSMP EC (20.1%). 28 patients were not classified due to the lack of results: sequencing for the *POLE* gene (6 patients) and lack of IHC results for mismatch repair (MMR) proteins (22 patients) (Figure 1). 

Table 1 displays the clinicopathological characteristics of the 139 patients who were included in our study. The current study encompassed patients distributed across various clinical stages: two in stage IA1, characterized by non-aggressive histological types either restricted to an endometrial polyp or confined solely to the endometrium; five in stage IA2, marked by non-aggressive histological types affecting less than half of the myometrium and exhibiting none or focal LVSI; and two patients categorized under stage IA3, featuring low-grade endometrioid carcinomas confined to the uterus. The remaining patients were diagnosed with stage IB (*n* = 58) and IC (*n* = 72) stages.

Disease recurrence occurred in 12 (8.6%) of the 139 patients. The median follow-up time at the time of data cut-off (1 June 2023) was 53.41 months (range: 26.24–76.58 months). The median time between surgery and recurrence was 19.30 months. Recurrences of the disease occurred in 12 (8.6%) women only in the clinical stage IC according to the 2023 FIGO staging system with aggressive forms of EC (EEC G3, Serous, and CCC) (Figure 2A,C). In 47 patients with stage I EC, according to the 2023 FIGO staging system with abnormal p53 IHC expression (p53abn), recurrence was demonstrated in 9 patients (19.1%) (Figure 2E). 

Based on the univariate logistic regression model of the patients’ clinical and pathological risk factors, there are three factors related to cancer recurrence. The independent risk factors are: stage IC (HR = 1.86, 95% CI 1.45–8.18, *p* < 0.001), aggressive histological types EC (HR = 5.08, 95% CI 1.79–7.03, *p* < 0.001), and the presence of p53abn protein immunoexpression (HR = 4.48, 95% CI 1.26–6.68, *p* = 0.009). 

In the multivariate analysis, the aforementioned three factors were also associated with recurrence. The independent risk factors are stage IC (HR = 3.01, 95% CI 1.97–6.94, *p* < 0.001), aggressive histological types EC (HR = 2.86, 95% CI 1.73–5.25, *p* < 0.001), and the presence of p53abn protein immunoexpression (HR = 3.49, 95% CI 1.11–6.34, *p* = 0.011) (Table 2).

During the 60-month follow-up period, 23 patients died. Table 3 summarizes the univariate and multivariate Cox proportional hazards models for OS. For OS, stage IC, aggressive histological types EC, and the presence of p53abn protein immunoexpression were all significantly associated with lower 5-year OS rates (Figure 2B,D,F). 

Based on the univariate analysis, the independent prognostic factors for OS are: stage IC (HR = 3.66, 95% CI 1.29–6.98, *p* = 0.018), aggressive histological types EC (HR = 4.78, 95% CI 1.46–6.82, *p* = 0.025) and the presence of p53abn protein immunoexpression (HR = 4.24 95% CI 1.15–6.44, *p* = 0.010). 

Similarly, the three factors mentioned above were also associated with OS in the multivariate analysis: stage IC (HR = 2.99, 95% CI 1.16–5.24, *p* = 0.021), aggressive histological types EC (HR = 2.64, 95% CI 1.63–5.11, *p* = 0.027), and the presence of p53abn protein immunoexpression (HR = 3.15, 95% CI 1.02–5.94, *p* = 0.021) (Table 3).

## 4. Discussion

In early clinical stages, EC is a heterogeneous group that varies in the risk of recurrence and mortality, depending on clinical and pathological risk factors such as age, FIGO stage, myometrial infiltration depth, degree of histological differentiation, LVSI, and histological type of cancer [16,17]. Because of the wide range of risk levels, the potential benefit of adjuvant therapy for each patient is also variable [2]. This awareness emphasizes the importance of individualizing treatment as much as possible, considering adjuvant therapy’s side effects. The risk of over- or under-treatment of some patients is still being debated. Limited data is available to assist decision-making about the extent of treatment [16,18,19]. There is also no consensus on the best method of adjuvant treatment for patients with early-stage non-endometrioid endometrial cancers, despite the literature indicating a high risk of disease recurrence at stage IA (FIGO 2009) [17,20].

In the 2023 FIGO staging system for EC, the IC stage is designated for aggressive histological types that are either contained within a polyp or restricted to the endometrium. This stage encompasses several varieties, including grade 3 endometrioid endometrial carcinoma (EEC G3), serous carcinoma (SC), clear cell carcinoma (CCC), mixed carcinoma (MC), uterine carcinosarcoma (UC), and carcinosarcoma (CS), as well as other less common types like mesonephric-like carcinomas and gastrointestinal mucinous type carcinomas [5].

It Is well understood that the histological picture does not always predict tumor behavior and the histological phenotype does not always correlate with genotype. Kandoth et al. discovered that 5% of G1/G2 EECs and 25% of G3 EECs had mutations similar to serous carcinomas (SC) and that they may behave more aggressively [11]. The risk of recurrence of non-endometrioid carcinomas in stage I, according to FIGO, indicated in various studies, ranges from 0 to 70%, with most authors stating that the risk in stage IA ranges from 10% to 25% [21,22,23]. 

EC confined to a polyp is a clinical course that offers particular difficulty given the rarity of disease and limited available published data. Given the limited extent of disease but the aggressive biologic behavior of low-grade ECC, SC, and CCC it is unclear whether adjuvant therapy is warranted. Recurrences have been reported even in a small number (9%) of patients, with disease confined only to a polyp and in 30% of patients with EC confined only to the endometrium [24,25,26]. Among histologically aggressive forms of EC, SC type accounts for less than 10% but is responsible for nearly 50% of recurrences and deaths [27]. The overall estimated 5-year overall survival rate for women diagnosed with SC is 55% [28]. 

The comprehensive surgical evaluation of SC staging is critical, as 37–39.4% of patients without myometrial invasion were found to have metastases [29,30]. Donkers et al. [31] analyzed the cases of 272 women with SC, among whom 44% were clinically stage I according to 2009 FIGO staging system. Overall, 48% of the subjects had a recurrence of the disease, with the majority (58%) having distant metastasis. Moreover, even in stage IA patients, a high recurrence rate (42%) was observed. 

Difficulties in clinical decision-making apply to patients with SC detected in preoperative biopsy, with disease confined to a polyp or the endometrium confirmed by tissue examination after hysterectomy. Hui et al. [32] observed no recurrences of disease in patients with disease confined to a polyp during a 26-month follow-up period. On the contrary, other authors analyzing histologically aggressive forms located in a polyp or the endometrium observed recurrences of the disease [26,33,34,35]. In our study, disease recurrence occurred in 12 (8.6%) of 139 patients with EC in the IC stage according to the 2023 FIGO staging system, with aggressive forms of EC (EEC G3, serous, and CCC).

The landmark Cancer Genome Atlas study performed an integrated analysis of the genome, transcriptome, and proteome of 373 EC samples and identified 4 molecular subtypes that naturally clustered into 4 profiling patterns differentiated by patient survival [11]. The FIGO guidelines (2023 edition) recommend incorporating molecular subtype (*POLE*mut, MMRd, NSMP, p53abn) into clinical staging criteria in all EC cases in order to stratify risk groups as potential factors influencing decisions on adjuvant or systemic treatment. 

An endometrial biopsy can be used to determine the molecular subtype. EEC with high malignancy (G3) is a heterogeneous disease in terms of prognostic, clinical, and molecular factors. This is the type of cancer that benefits most from using molecular classification to improve prognosis and treatment decisions [36]. Without molecular classification, EEC with high malignancy cannot be adequately assigned to a risk group, and, therefore, molecular profiling is especially recommended for these patients. For practical reasons, EECs with high malignancy are grouped with aggressive histological types in the current FIGO classification to avoid inadequate treatment coverage (if molecular classification is unknown) [5]. 

The WHO has officially recognized a definitive molecular categorization for EC, which distinctly classifies p53-abnormal (p53abn) EC as a subgroup. This classification is clinically relevant, as it denotes the subgroup with the most adverse prognosis. Moreover, patients within this category are often identified as the most suitable candidates for receiving combined adjuvant chemotherapy and radiation therapy, given their prognostic profile. Endometrial cancers with mutations in the *P53* gene are overwhelmingly classified as high-risk cancers based on traditional histopathological evaluation [37]. However, in some patients with tumors classified as low/intermediate risk based on histopathological examination, the diagnosis of a *P53* mutation will result in a change in the classification and the emergence of indications for adjuvant therapy [4].

In a study conducted by Talhouk et al. [38], tumors with abnormal p53 protein were G3 tumors in 96% of cases, non-endometrial tumors in 80.2% of cases, LVSI infiltration was found in 61% of cases, and had a high rate of lymph node metastasis. According to Jamieson et al. [39], lymph node metastases were observed in 45% of cancers with the *P53* mutation. Iwai et al. [40] claim that p53 abnormalities may not be detected in preoperative samples because preoperative biopsies performed in their study did not show abnormal p53 IHC staining; however, they demonstrated p53 overexpression in 56% of cancers in hysterectomy specimens. In our study, among patients with EC at clinical stage I according to the 2023 FIGO staging system, IHC p53abn was found in 33.8% (47), IHC MMRd in 20.9% (29), *POLE* gene mutations—*POLE*mut in 25.2% (35) and IHC NSMP in 20.1% (28) of patients. Among 47 patients with EC stage I, according to the 2023 FIGO staging system with abnormal p53abn IHC (p53abn), recurrence occurred in 9 (19.1%) patients. 

In their study, Ouyang et al. [41] conducted an in-depth analysis of the clinicopathological characteristics of patients diagnosed with stage 1A uterine serous carcinoma (USC), focusing on cases both with and without the involvement of polyps. Additionally, they evaluated the impact of factors such as the size of the polyp and the presence of LVSI on these patients. Their findings assert that, for patients classified under stage 1, there is no significant variation in overall survival rates, irrespective of whether they underwent active treatment or were under observation. This observation holds across the spectrum of clinical interventions and surveillance strategies applied to this patient group. On the other hand, Assem et al. [42] show a trend of significance for worse OS in non-polyp serous EC-confined patients. Barlin et al. [43] showed that grade, EC histological subtype, and staging according to the 1988 and 2009 FIGO staging classification were all significantly associated with OS. The 5-year OS rates for the 1988 staging system were as follows in their study: IA, 95.4%; IB, 88.4%; and IC, 81.1% (*p* < 0.001). The 5-year OS rates for the 2009 staging system were IA, 91.5%; and IB, 81.1% (*p* < 0.001). In our research studies, patients with clinical stage IA, according to the 2023 FIGO staging system, had a 5-year survival rate of 93.4%, while patients with IC had a 5-year survival rate of 77.1% (*p* = 0.018). It was also demonstrated that histological type also affected OS.

## 5. Conclusions

The findings from our research investigations corroborate the premise that within the clinical stage I endometrial cancer (EC), as per the recently updated 2023 FIGO classification, distinct prognostic outcomes are inherently tied to specific molecular subtypes. Notably, cases exhibiting *POLE* mutations (*POLE*mut) demonstrate the most favorable prognostic outcomes. Conversely, cases characterized by p53 abnormalities (p53abn) are associated with the least favorable prognoses. This delineation extends beyond the conventional clinicopathological evaluations previously employed. The introduction of the novel EC staging system by FIGO marks a significant advancement in our capability to accurately assess and categorize risks specifically for IC stage EC, enhancing the precision of prognostic stratification and potentially guiding more tailored therapeutic approaches.

## Figures and Tables

**Figure 1 cancers-16-00390-f001:**
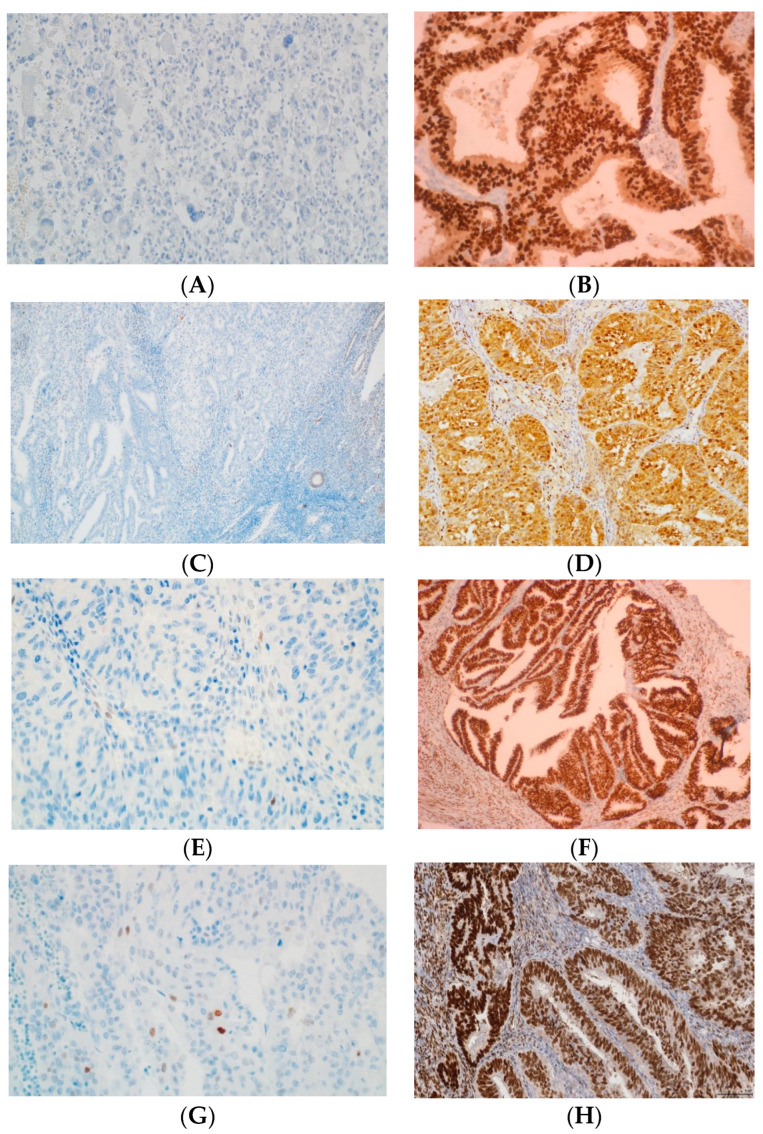
Examples of p53, MLH1, MSH2, MSH6, and PMS2 IHC in EC. (**A**) Negative p53 staining and (**B**) positive of p53 staining in tumor glands (mutant overexpression). (**C**) Negative MLH1 staining and (**D**) positive of MLH1 staining in tumor glands. (**E**) Negative MLH2 staining and (**F**) positive of MLH2 staining in tumor glands. (**G**) Negative MSH6 staining and (**H**) positive of MSH6 staining in tumor glands. (**I**) Absence PMS2 staining and (**J**) positive of PMS2 staining in tumor glands. All original images are captured at a magnification of 200×.

**Figure 2 cancers-16-00390-f002:**
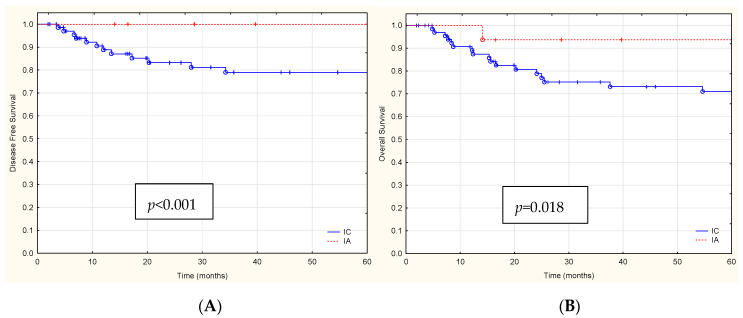
**The** Kaplan-Meier survival analysis (**A**) DFS and (**B**) OS for FIGO stage IA and IC patients (red and blue lines), (**C**) DFS and (**D**) OS for patients with aggressive histological EC types G3, Serous, CCC vs. G1/G2 (red and blue lines), (**E**) DFS and (**F**) OS for patients with p53abn vs. *POLE*mut (red and blue lines).

**Table 1 cancers-16-00390-t001:** Clinicopathologic characteristics according to the four molecular subgroups in stage I EC.

	Total	p53abn	MMRd	*POLE*mut	NSMP
	*n* = 139 (100%)	*n* = 47 (33.8%)	*n* = 29 (20.9%)	*n* = 35 (25.2%)	*n* = 28 (20.1%)
**Age, years**					
<60	58 (41.7)	5 (10.6)	10 (34.5)	24 (68.6)	19 (67.9)
≥60	81 (58.3)	42 (89.4)	19 (65.5)	11 (31.4)	9 (32.1)
**FIGO**					
IA	9 (6.5)	0 (0)	0 (0)	9 (25.7)	0 (0)
IB	58 (41.7)	1 (2.1)	15 (51.7)	26 (74.3)	16 (57.1)
IC	72 (51.8)	46 (97.9)	14 (48.3)	0 (0)	12 (42.9)
**Histological subtype**					
EEC G1/G2	67 (48.2)	0 (0)	19 (65.5)	34 (97.1)	14 (50.0)
EEC G3	40 (28.8)	23 (48.9)	7 (24.1)	1 (2.9)	9 (32.1)
Serous	24 (17.4)	18 (38.3)	2 (6.9)	0 (0)	4 (14.3)
CCC	8 (5.6)	6 (12.8)	1 (3.5)	0 (0)	1 (3.6)
**Myometrial invasion**					
<50%	81	46 (97.9)	9 (31.0)	16 (45.7)	10 (35.7)
≥50%	58	1 (2.1)	20 (69.0)	19 (54.3)	18 (64.3)
**LVSI**					
None	112 (80.6)	42 (89.4)	21 (72.4)	33 (94.3)	16 (57.1)
Focal	27 (19.4)	5 (10.6)	8 (27.6)	2 (5.7)	12 (42.9)

**Table 2 cancers-16-00390-t002:** Cox proportional hazard model Univariate and multivariate analyses for disease-free survival (DFS).

	Univariate Analysis			Multivariate Analysis		
	HR	95% CI	*p*	HR	95% CI	*p*
**Age, years**						
<60	1			-	-	-
≥60	1.31	0.73–2.51	0.171	-	-	-
**FIGO**						
IA	1			1		
IB	1.03	0.64–2.11	0.245	-	-	-
IC	3.86	1.45–8.18	<0.001	3.01	1.97–6.94	<0.001
**Histological subtype**						
EEC G1/G2	1			1		
ECC G3, Serous, CCC	5.08	1.79–7.03	<0.001	2.86	1.73–5.25	<0.001
**Myometrial invasion**						
<50%	1			-	-	-
≥50%	1.57	0.89–2.47	0.158	-	-	-
**LVSI**						
None	1			1		
Focal	1.44	0.54–2.03	0.161	1.16	1.06–1.99	0.302
**Molecular subgroups**						
*POLE*mut	1			-	-	-
NSMP	0.84	0.32–1.34	0.765	-	-	-
MMRd	0.94	0.52–1.91	0.764	-	-	-
p53abn	4.48	1.26–6.68	0.009	3.49	1.11–6.34	0.011

**Table 3 cancers-16-00390-t003:** Cox proportional hazard model Univariate and multivariate analyses for overall survival (OS).

	Univariate Analysis					Multivariate Analysis
	HR	95% CI	*p*	HR	95% CI	*p*
**Age, years**						
<60	1			-	-	-
≥60	1.12	0.65–2.04	0.167	-	-	-
**FIGO**						
IA	1			1		
IB	1.02	0.61–1.98	0.177	-	-	-
IC	3.66	1.29–6.98	0.018	2.99	1.16–5.24	0.021
**Histological subtype**						
EEC G1/G2	1			1		
ECC G3, Serous, CCC	4.78	1.46–6.82	0.025	2.64	1.63–5.11	0.027
**Myometrial invasion**						
<50%	1			-	-	-
≥50%	1.31	0.71–2.31	0.235	-	-	-
**LVSI**						
Absent	1			1		
Focal	2.21	1.64–3.11	0.062	2.04	1.24–3.06	0.073
**Molecular subgroups**						
*POLE*mut	1			-	-	-
NSMP	0.78	0.30–1.27	0.365	-	-	-
MMRd	0.84	0.48–1.78	0.524	-	-	-
p53abn	4.24	1.15–6.44	0.010	3.15	1.02–5.94	0.021

## Data Availability

All data are available on requested.

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
