# Peer review of "Prognosis of Stage I Endometrial Cancer According to the FIGO 2023 Classification Taking into Account Molecular Changes"

_cancers, 2024, doi:10.3390/cancers16020390_

Round 1

Reviewer 1 Report

Comments and Suggestions for Authors

            After a thorough and repeat consideration of the study by Dr. Dobrzycka and colleagues, I strongly recommend its prompt acceptance and publishing, provided minor stylistical and formatting imperfections of the text are corrected (see below). 

            Their MS offers new data important for a more individualized approach to molecular diagnosis and subsequent treatment of stage I endometrial cancer. These results were based on a rather extensive material derived from 167 endometrial cancers of which 139 yielded accurate molecular testing. The final results stem from in-depth statistical analysis, are highly interesting and with a possible direct translation into clinical practice.  

            The work has a proper structure and is very well written, the medical English language being fully proficient.  

            Minor issues include:  

- I am not sure whether the abbreviation EEC was explained with its first use;

- Page 1, is: ‘Stage I (IA1): A non-aggressive histologic type of EC … ’; let me humbly suggest a clarification: ‘Stage I: IA1 is a non-aggressive histologic type of EC … ’;

- Page 1, last line, is: ‘50% of the myometrium … ’; should read: ‘less than 50% of the myometrium … ’; 

- Page 2, line 2, is: ‘ … involving 50% of the myometrium … ’; should read: ‘… involving 50% or more of the myometrium … ’; 

- Page 3,  the subchapter ‘2.2.2. POLE gene mutation analysis’: the first phrase of this subchapter requires grammar correction; also, pls arrange more order to the abbreviation PCR; ‘The list of primer sequences ... ’ requires ‘is’; pls explain the abbreviation: dNTPs;  

- Page 3, a link given to the Primer3Plus software: https://www.promer3plus.com seems to contain an important lettering; 

- Page 4, line 18 from above: consider replacing ‘discovered’ by ‘detected’;  

- Page 8, pls correct: ‘[15, 15]; 

- Page 9, first lines, correct: ‘The risk of implementing over- or under-treatment of some patients in the early clinical stages of EC The risk of over- or under-treatment of some patients is still being debated.’;  

- References 10, 21 and 37 are identical; pls leave # 10 only and adapt citations in the text accordingly; 

- References #2, 3, 5, 7, 9, 18, 20, 32, 35, 38, and 43 have their titles unnecessarily capitalized.  

Comments on the Quality of English Language

As given above.

Reviewer 2 Report

Comments and Suggestions for Authors

The study addresses a critical aspect about endometrial cancer prognosis depending upon molecular classification. However, following revision deemed necessary to make it suitable for publication.

1. The abstract need to be re-written, as it is bit confusing and not clear. The message in the abstract needs to be in clear and simple language.

2. It will be better if the authors add immunohistochemistry images as a figure for better assessment.

3. It will be better if the authors add more details about endometrial cancer along with all the subtype names in the introduction section, so, readers from different background can have a better picture about this disease.

4. Authors need to provide the results in more details along with subheading for each parts.

Reviewer 3 Report

Comments and Suggestions for Authors

This is an interesting study, providing validation of the new FIGO 2023.

The study is interesting, but the results are presented in a somewhat confusing way, and also the discussion can be improved. 

My main concern is tha appropriate use of FIGO 2023. Table 1 shows 72 tumors stage IC (agressive histological types limited to a polyp or confined to the endometrium) , while there were 9 cases stage IA (IA1, non-agressive histological types limited to an endometrial polyp of confined to the endometrium and IA2 non-aggressive histological types involving less than 50% of myometrial wall, with no or focal LVSI), and 58 stage IB cases (non-aggressive histological types involving more than 50% of myometrial wall and with no or focal LVSI).  These results are very unusual, since aggressuive histological trypes limited to a polyp or confined to the endometrium are usually not that frequent. I am questioning whether some of these stage IC cases are really having myometrial involvement and are really stage IIC.

The number of tumors with less than 50% myometrial invasion is 80, and all of cases stage IA, and IC count 81.

I suggest the authors to re-check the number of stage IC cases, and to include the whole stage groups, IA1, IA2, IA3, IB and IC.

Without clarifying this, it is difficult to evaluate the paper. 

Comments on the Quality of English Language

English grammar could be improved
